# Secondary School Athletic Trainers’ Navigation of Patient Socioeconomic Status Challenges in Care: A Qualitative Study

**DOI:** 10.3390/ijerph192416709

**Published:** 2022-12-13

**Authors:** Mayrena Isamar Hernandez, Elena Catherine Miller, Kevin Mark Biese, Luis Columna, Susan J. Andreae, Timothy McGuine, Traci Snedden, Lindsey Eberman, David Robert Bell

**Affiliations:** 1Department of Kinesiology, Sam Houston State University, Huntsville, TX 77340, USA; 2Department of Kinesiology, University of Wisconsin–Madison, Madison, WI 53705, USA; 3Department of Kinesiology, University of Wisconsin–Oshkosh, Menasha, WI 54952, USA; 4Department of Orthopedics & Rehabilitation, University of Wisconsin–Madison, Madison, WI 53705, USA; 5School of Nursing, University of Wisconsin–Madison, Madison, WI 53705, USA; 6Applied Medicine and Rehabilitation, Indiana State University, Terre Haute, IN 47803, USA

**Keywords:** socioeconomic status, social determinants of health, patient-centered care, health disparities

## Abstract

Secondary school ATs (SSATs) are uniquely positioned healthcare providers at an optimal public health intersection where they can provide equitable healthcare to low socioeconomic status (SES) adolescents. SSATs face similar challenges to physicians in treating low SES patients, but their strategies may be different compared to other medical professions. However, the consequences of low SES population healthcare delivery by SSATs have not been explored. SSATs were asked to share what challenges, if any, they encounter with providing care for their low SES patients and what strategies they find most effective to overcome these challenges. Data were collected via semi-structured interviews and reflective field notes and analyzed using a four-step, interpretative phenomenological analysis (IPA) guided theme development. Data saturation was met, and the sample size aligned with other IPA studies. Trustworthiness was established with research triangulation and Yardley’s four principles. Three interrelated themes emerged: (a) mechanisms for identifying SES, (b) the impact of SES on care, and (c) navigating SES challenges. SSATs described many strategies that were gained through their clinical experiences to overcome healthcare barriers. SSATs have the potential to decrease health disparities through their role as a liaison and advocates for their low SES patients.

## 1. Introduction

There is substantial evidence that socioeconomic status (SES) affects an individual’s health outcomes and the health care they receive. Some examples of health disparities for the low SES population include worse self-reported health, lower life expectancy, and suffer from more chronic conditions, and limited access to health care as compared to high SES populations [1,2]. Previous research has demonstrated that compared with other patients, physicians’ perceptions of low SES patients have impacted clinical decisions [3,4]. Physicians accommodate their management plan to suit those with financial difficulties, public/no insurance, and lower health literacy in an attempt to aid low SES patients [4]. However, these changes can inadvertently lead to patients receiving less than ideal or non-standard treatment, such as less aggressive management and/or postponing testing, more generic medications, and avoiding referral to specialty care which leads to worse outcomes [2,3,4,5]. Many of these less-than-ideal clinical decisions leave physicians feeling helpless and frustrated when faced with the complexity of SES and its intertwined social determinants of the health of their patients [1].

The field of medicine has historically operated under a “downstream” approach paradigm. Meaning many individuals do not receive care until there is a disease or injury has occurred, demonstrating only a secondary or tertiary level of prevention [6,7]. However, this approach has not proven to be effective with patients who are of low SES due to delayed interventions and access to health care. Due to the strong evidence of the negative impact of low SES on health and health, there is a public health priority in the healthcare system to reduce disparities through an “upstream” approach through a primary level of prevention [7]. Physicians and many other allied health professionals already engage in a wide range of clinical preventative practices with the aim of preventing disease and promoting lifelong health [1,8]. Specifically, to athletic trainers (ATs), the health care they provide is at the tertiary level of prevention through rehabilitation and return to sport, a secondary level of prevention by evaluation and diagnosis of injuries and medical conditions, and a primary level of prevention through the pre-rehabilitation program and set protocols for sports participation to mitigate injuries or medical condition from occurring [7,9,10]. 

Secondary school ATs (SSATs) are uniquely positioned healthcare providers at an optimal public health intersection where they can provide equitable healthcare to vulnerable low SES adolescents [9]. ATs are essential in providing a high standard of care which impacts life-long health and physical activity during a critical time such as adolescence. However, the consequences of low SES population health and healthcare delivery by ATs have not been explored. ATs provide direct care to a significant number of low-SES students attending public secondary schools [11]. Post et al. observed that nearly 95% of all secondary schools in their study used AT healthcare services in some capacity, such as medical coverage and preventative services [12]. For low SES student-athletes, ATs in the secondary school setting might be one of their primary forms of accessible health care.

Barter et al. demonstrated that there are socioeconomic disparities in access to ATs in the public secondary school setting [13]. Similarly, Robison et al. found that schools of lower SES had fewer contact frequencies for injury-related care, yet, equal rates of therapeutic exercise, neuromuscular control, and manual therapy were achieved [14]. These findings show that ATs may face similar shortcomings alongside physicians when providing care to their low SES patients once advanced orthopedic consultation, imaging, and advanced testing are needed [4]. However, because ATs are in direct contact with patients in the secondary school setting, the challenges and strategies in caring for low SES patients may be different compared to previous research done in other medical professions. Therefore, the purpose of our study was to explore the strategies secondary school ATs implement to overcome the challenges related to providing care to the low SES population. Furthermore, we will investigate how AT’s education and clinical experiences prepared them to overcome these barriers they may face when providing care to low SES patients. This is important because understanding current strategies or challenges to overcome barriers related to low SES patient care can help prepare current and future athletic trainers in the secondary school setting to uphold a standard of care and consider the social determinants of health (SDOH) of their patients.

## 2. Materials and Methods

### 2.1. Research Design

To further elucidate an understanding of AT’s clinical management decisions toward their low SES student-athletes, this study utilized an interpretative phenomenological analysis (IPA) research approach [15]. This qualitative research approach has theoretical roots drawn from phenomenology, hermeneutics, and idiography. IPA represents the phenomenological method in that it is primarily concerned with examining each individual’s experiential account versus an objective description of an event. IPA’s roots in hermeneutics are demonstrated by this research approach being an interpretative endeavor, where the researchers “make sense of the participant making sense” of their embodied experiences as a secondary school AT providing care for low SES individuals [16]. Lastly, IPA is idiographic through its concern with understanding the experience of each individual participant in detail [17]. The goal of this qualitative inquiry was to distinguish the experiences and extricate emergent themes and patterns to enrich the understanding of the role of a secondary school AT in providing care to low SES student-athletes. The XX Review Board reviewed and approved the qualitative study protocols.

### 2.2. Procedures and Instrumentation

A specific qualitative interview protocol was developed (Table 1). Due to the lack of a pre-existing instrument, the research team developed an interview protocol guided by the aims research questions. The primary research questions included the following: (1) What are, if any, the challenges secondary school ATs face when providing care to their low SES student-athletes? (2) Under what patient circumstances, if any, do secondary school ATs encounter difficulty with providing care for their low SES student-athletes? (3) What strategies do secondary school ATs find most effective when providing care for their low SES student-athletes?

The semi-structured interview protocol was developed to include 11 open-ended questions pertaining to the participants’ experiences, challenges, and strategies for providing care for their low SES student-athletes. The semi-structured nature of the interview script allows for flexibility to ask clarifying questions that could potentially lead to new topics not previously addressed. When the interview protocol was developed, it was reviewed by content experts to ensure face validity. Prior to the commencement of the data collection, the interview protocol was pilot tested with 3 individuals who met the inclusion criteria of being a secondary school AT but who were not participants during data collection. The purpose of these pilot interviews was to prepare the interviewer (PI) and confirm the comprehensiveness of the interview script. Based on the pilot interviews and participant feedback, the interview questions were recorded and/or modified. To ensure consistency across interviews, the PI conducted all the Zoom interviews for this aim. After completing the survey of the initial study conducted by Hernandez et al., participants were asked to provide their email addresses if they wanted to participate in a future qualitative inquiry [18]. An email explaining the purpose of the qualitative portion and an invitation to participate in the study was sent to all those individuals who expressed interest. Voluntary written consent was implied when interested individuals responded to the researcher and indicated that they wanted to participate. When the individual agreed to participate, a 60-min interview was scheduled. Prior to the start of each interview, the participant was asked to provide verbal consent for the interview to be digitally recorded via Zoom. At the time, the PI identified any biases by explicitly publicizing their positionality to the participant. Once the interview was complete, audio was extracted and sent to a third-party transcription service. Transcriptions of the interviews were sent to the participant to ensure the validity of the data through member checking. In member checking, the participant was allowed to review their transcript to confirm the data was transcribed correctly and allow for any clarification or removal of data [19]. Each interview was blinded, and participants were given a pseudonym.

### 2.3. Participants and Sampling

From the initial cross-sectional survey from Hernandez et al. [18], 139 secondary school ATs (37% of the study population) expressed interest in completing an interview with the research team. Due to the exploratory nature of qualitative research and IPA, approximately 12 participants were needed to reach data saturation [15,16,20]. Data saturation occurs when the interviewer no longer obtains new information from the participants and sees a redundancy in the data [21,22,23]. Participants’ demographics, including the highest level of education, race/ethnicity, years of AT clinical experience, secondary school setting, title 1 status, school locale, and pseudonyms are provided in Table 2.

### 2.4. Data Analysis

Data were analyzed thematically using a four-step IPA analytical process [21,24]. The objective of this process was to capture and present the results in the form of participants’ embodied experiences. In the first step, the investigators read and reread and/or will listen to each participant’s transcript interview and related field notes several times to develop a deep understanding and familiarity with each participant and implement multiple-analyst triangulation [20]. While reading and rereading, and/or listening, the investigators will note items of interest and early interpretative commentary in the transcripts and field notes in the form of descriptive and exploratory comments. Second, the investigators reduced transcripts, reflective notes, and descriptive exploratory comments associated with each case into emergent experiential grounded themes and met to compare notes and come to a consensus. During this meeting, the team created the initial codebook by discussing their respective themes and conceptualizing the core ideas. At this stage, themes will reflect both the participant’s words as well as the authors’ interpretation of those words. The codebook was audited by an external reviewer, and the consensus codebook was confirmed [25]. In the third step, emergent themes were compared within each participant’s documents to form a set of inductive clusters or related themes. Throughout this process, all steps were completed for each participant’s data independently at the case level. After thematic clusters are identified at the case level, the final step is to search for patterns and connections across participants through constant comparison. The investigators reviewed the themes with the rest of the research team to ensure that they were in line with the purpose and framework of the study. Thematic clusters that were considered in line with the purpose and framework of the study were summarized and presented as results.

Yardley’s four principles for assessing the quality of the qualitative research for use in IPA studies were followed to evaluate this research study [17,24]. These four principles include (a) sensitivity to context, (b) commitment and rigor, (c) transparency and coherence, and (d) impact and importance. Sensitivity to context was considered by the principal investigator, explicitly publicizing their positionality as a researcher, certified and licensed AT, and a previous low SES youth athlete to participants to uncover any potential biases during the interviews. The participants’ voices were demonstrated using an abundant number of verbatim transcript quotes in the results to allow readers to check interpretations. The commitment was supported by inviting participants to review their original transcriptions to correct any misrepresentations, elaborate, or delete content if desired. Participants were not asked to review interpretations of themes as this is incongruent with the generations of data. Rigor or the completeness of the data collection and analysis were supported by utilizing an interview guide that was developed by existing literature in physician clinical decision management and focus on the AT secondary school setting [4,26]. Transparency was achieved by explicitly describing the research process (recruitment, interview, transcription, and analytic procedure, accounting for research positionality, reflexivity, and bias). Coherence between the research questions and research approach was supported by the value of phenomenological research in explicating lived experiences of the participants in this study. Lastly, the impact and importance of this qualitative research were achieved in the ability of authors to communicate the content as clinically applicable and useful. This impact and importance were ultimately judged by the readers consuming this study [24,27].

## 3. Results

Three interrelated themes and subsequent subthemes emerged from the interviews (Figure 1): mechanisms for identifying SES, the impact of SES on care, and navigating SES challenges in care.

Mechanisms for identifying SES describe how participants define SES through characteristics and assumptions of the low and high SES populations (Table 3). These definitions come from participants’ lived experiences with a low SES population and can be biased based on these experiences. Within the mechanisms for identifying SES theme, ATs described several strategies to identify low SES patients at the secondary school, which was defined as a plan, method, or series of maneuvers for obtaining a classification of a person’s SES. This was displayed through methods of observation such as noting methods of transportation their family used or which zip code their patient’s family resided in. ATs also used their patient’s insurance status and information about their patient’s SDOH through communicating with youth sports stakeholders and their patients. Information gathered allowed us to identify patients who would be of low SES and present challenges in the delivery of AT care. However, their preparation through professional education varied. Preparation was defined as a reflection on the AT’s athletic training program (ATP) (classroom and/or clinical) and how it prepared these secondary school ATs to identify low SES patients. Many ATs described not understanding the mechanisms for identifying SES until a few years into their career and not being taught about it in their classroom (Table 3).

The impact of SES on care was defined as an obstacle(s) that prevents progress in the health care delivered to low SES students (Table 4). This theme was further supported by barriers to in-house care, which was defined as barriers to health care that was provided in the athletic training secondary school setting and provided by the athletic trainer. This was described through limitations based on the location of the secondary school impacting resources for providing care to student athletes, athlete’s financial limitations impacting appropriate equipment for sports participation, language barriers of parent/guardian, and non-compliant patient/guardian due to household dynamic. (Table 4).

Another subtheme that emerged in the impact of SES on care was the barriers to the health care system, which was defined as lack of insurance or a limited option to health care services through public insurance, evaluation from doctors, and time to surgery. This was described by participants as their low SES patient’s geographic location impacting their built environment, such as the quality of hospitals and clinics accessible, transportation and insurance limitations on access to health care, and institutional distrust of the health care system (Table 4). ATs described this as being difficult to navigate when their patients compare themselves to peers who are receiving outside care/referrals quickly due to their insurance status and SDOH differences. Many of the ATs described this as an everyday reality in their health care practice which leads to difficult conversations and frustration from lack of resources for their health care practice and for their patients.

Participants noted the need to navigate SES challenges in care, which was defined as a plan of action, methods, and use of resources to achieve a certain desired healthcare goal for their low SES patients (Table 5). This theme is further supported by subthemes of (1) liaison, (2) developing rapport, (3) and athletic training program (ATP) experiences. Liaison is defined as a health care provider who works closely with doctors, health insurance administration, school administration, youth sports parents, and community stakeholders. The role of liaison was described by participants as a method to avoid delays in health care by identifying their patients of low SES and then finding resources and advocating for their low SES patients (Table 5). Developing rapport includes not only the relationship as a health care liaison but also stresses the importance of developing relationships over time with the students and their parents/guardians to gain their trust as a health care provider, which yields timely identification of their patients of low SES and finding resources/advocating for them (Table 5).

ATP experiences for navigating SES challenges in care is a reflection on the AT’s ATP (classroom and/or clinical education) and how it prepared these secondary school ATs to provide care for their low SES patients. Many ATs indicated specific clinical experiences in low SES settings that helped them learn about the impact of patient SES and how to navigate the SES challenges in clinical management decisions. These low SES clinical secondary school sites were described as valuable for ATs who are currently in the secondary school setting as compared to only having a collegiate sport setting as their clinical site through their ATP (Table 5).

## 4. Discussion

This study focused on secondary school AT’s experiences with providing care to the low SES patient population. Our study follows up on a previous study focused on describing secondary school AT’s perceptions of the SDOH and how their patient’s SES can impact their clinical management decision when it pertains to referral for advanced care [18]. Our study allowed us to explore valuable information based on clinical experiences as certified ATs providing care to low SES patients in the secondary school setting. The most important finding of this study was that ATs experience many challenges with providing care to low SES patients in the secondary school setting. In addition, our study demonstrated that ATs face similar challenges to physicians in treating low SES patients [3,4,28]. However, because ATs are in direct contact with patients in the secondary school setting, their strategies in caring for low SES were different compared to previous research done in other medical professions. Literature has demonstrated that the SDOH influences patient health and health care [29,30,31]. The ATs in our study were able to reduce the influence of the SDOH on their patients through awareness of their impact on health outcomes and their strategies to intervene and navigate the challenges associated with their low SES patients. ATs need to be more aware of the SDOH due to its complexity of rarely being a single negative SDOH negatively impacting health, especially in the lives of their low SES patients [30,32]. Our study is the first to our knowledge to provide qualitative findings on secondary school ATs perceptions, challenges, and, most importantly, strategies for navigating clinical care in their low SES patient populations.

### 4.1. Mechanisms for Identifying Socioeconomic Status

Understanding the SDOH can help ATs better target their patient outreach and engagement efforts by identifying patients who need more community support and social services to overcome barriers to health care [7]. The first step in addressing hidden socioeconomic issues as a health care provider is identifying potential social challenges of their patients in a sensitive and culturally acceptable, and caring way [1]. There are a growing number of clinical tools that have been created to help health care providers ask their patients about social issues such as lack of employment, food insecurity, discrimination, taboo topics such as abuse and trauma, and other issues such as low health literacy, legal or immigration status, and distrust of the health care system but none of these tools have been validated for the secondary school setting or with adolescents [33,34,35,36]. ATs in our study relied on many self-taught methods of identifying low SES patients in their secondary school setting. Particularly with observation, ATs relied on various aspects of the social determinants of health (SDOH) to indicate a patient’s SES. The SDOH are defined as the environments where people grow, work, and live and the broader set of forces and systems that influence their lives [29]. These forces can include political and economic policies and systems, social policies and norms, and societal institutions. On the individual level, the SDOH appears as housing, employment status, and working conditions [30].

Secondary school ATs are in a unique position in which they are able to see an intersection of the SODH in their adolescent patient population. In the case of our participants, ATs were able to identify the SES of their patients’ strategies they learned through ATC experience or through their clinical education in their ATP. These strategies revolved around documentation of their patient’s insurance status, particularly in pre-participation examinations, with public or non-insured patients being associated with low SES and health care disparities. Furthermore, ATs were able to use their observation skills by noting what method of transportation their patient took to get home after school. Specifically, patients lacked transportation or had to use public transportation as compared to their more affluent peers.

ATs also used their skill set as a liaison to have conversations with youth sports stakeholders or develop rapport with the patient themselves to identify the SDOH of patients. The key indicators for these ATs were based on their patient’s housing and its geographic location, free-reduced lunch status, parent/guardian employment status and marital status, which have all been shown to be associated with low SES and health and health care disparities [7]. Failure to evaluate a patient’s SDOH and lack of awareness of their importance in healthcare interactions can result in the hindered ability to provide culturally proficient comprehensive patient-centered care and promote patient health and well-being [32]. ATs in our study stressed the need to identify their patient’s social challenges and SES in a sensitive, caring way to provide an upstream healthcare approach. In a study involving a survey of patient perceptions on health care, more than 40% of patients reported that their family doctor was unaware of their struggles related to the SES and SDOH [37]. Therefore, recent clinical guidance has encouraged healthcare professionals to have an augmented awareness of clinical flags and patient cues through observation, as well as incorporate social history questions into patient encounters [1]. Previous studies have shown that physicians that know how to ask about their patient’s SDOH are more likely to report helping their patients through these issues [8].

Despite the need for evaluation and awareness of the SDOH, the majority of the ATs in our study indicated they did not feel prepared by their ATP to identify low SES patients. ATs reflected that there was no formal classroom education about the SDOH but that some ATs were able to learn about the impact of the SDOH on patient health through their clinical education. This is a similar feeling for other clinicians across various health professions [30]. In a study involving family doctors and nurse practitioners, 88% of participants agreed that healthcare workers are at the frontline to address underlying social issues of their patients, yet only one-third had specific ways of asking their patients about these potentially sensitive topics [8]. There is evidence that compassion and empathy allow the development of rapport with patients to identify social issues and SES challenges, yielding more accurate diagnoses and plan of care [1,8,35]. For example, a simple screening tool developed by Brcic et al. asked patients, “do you ever have difficulty making ends meet at the end of the month?” was found to be 98% sensitive and 64% specific for identifying their patient’s SES based on living below the poverty line [36]. Future research should investigate how these tools function for the secondary school setting and ATs. Integrating crucial SES information into medical records can be helpful in ensuring that athletic trainers and secondary school youth sports stakeholders can take these into consideration when developing a plan of care. Furthermore, the Commission on Accreditations of Athletic Training Education recently updated its 2020 Standards for Accreditation of Professional Athletic Training Programs to include the SDOH [32]. In doing this, future generations of ATs can understand their impact on patients and thus influence patient health outcomes positively.

### 4.2. Impact of Socioeconomic Status on Care

Secondary school ATs in our study shared many experiences where they witnessed the impact of SES on the care they delivered or referred for their patient. Lower SES has been associated with less access to orthopedic physician appointments based on insurance status, longer wait times, and poorer outcomes for elective procedures [38]. Picha et al. demonstrated that ATs face many of the same shortcomings as physicians. This may be due to these perceptions highlighting the care that is provided when having to access advanced care such as imaging and surgeries [32]. There were also no different differences by school SES on the number of conservative care [14]. A unique aspect of the AT profession is that ATs not only witness potential healthcare disparities of patients when there is a need to access the healthcare system, but there are barriers to the delivery of healthcare that occur within their own secondary school athletic training rooms. Many ATs stated that at the beginning of their AT careers, they would not realize a patient’s SES until the evaluation or referral process and sometimes, if they were lucky, during the pre-participation examination documents. Once an AT was able to identify a low SES patient, this allowed them to have an awareness of the complex and interrelated SDOH conditions that impacts their patients. ATs in our study described many experiences of clinical decision barriers related to their patient’s SES through patient interactions in in-house care (in the secondary school) and having to access the health care system. Many of these barriers to health care were directly related to their patient and/or patient’s parent/guardian’s health literacy, primary language, transportation, education level, employment status, income and wealth, housing, public safety, food security, neighborhood environment, and social environment [39]. Furthermore, ATs in low SES secondary schools primarily driven by geographic location in rural areas described limited resources in their athletic training rooms and further distances from stores, hospitals, and clinics that would have medical resources for them to provide care to all of their patients. These limitations based on school SES aligns with a previous study that identified the differences reported in AT care were related to costs with strapping and modalities with more affluent secondary school having access to STIM and ultrasound machines compared to less affluent schools [14]. Future studies should investigate the continuum of care for low SES schools impacted by rural settings versus low SES schools in higher urbanized areas that might have more opportunities to lessen the negative impact of low SES by accessing advanced care and resources for their AT rooms.

### 4.3. Navigating Socioeconomic Status Challenges in Care

Once a low SES patient was identified, referral for advanced care was impacted, and reliance on conservative treatment or measures before referral for advanced care was preferred, as described by ATs in our study. This might demonstrate the nature of “in-house” medical care the AT profession is prepared to provide, such as acute/sub-acute, chronic, preventative, and emergency medical care within our scope of practice. However, the type of “in-house” care of each of these domains is affected by the SES of their secondary school. A previous study demonstrated the most utilized service for affluent SES schools was strapping services; on average, SES schools were modalities, and in disadvantages, SES was therapeutic exercises [14]. Our findings continue to strengthen the AT services characteristics toward low SES populations.

The clinical decision on which doctor to refer to for advanced care was of concern when providing care to students of low SES in the secondary school setting. When interviewed, ATs explained that their skill set as a liaison allowed them to make relationships with doctors who would be willing to see their low SES patients for free, at sooner times, and provide equitable health care. Furthermore, ATs in our study were able to act as a liaison by connecting their patient and their parent/guardian with resources for public or state insurance, refugee services, language translation services, free-reduced lunch applications, and equipment for safe sport participation and return to sport. Finding these resources for their patient was done so by collaborating and networking with coaches, school administration, community stakeholders, local hospitals, doctors, nurses, and other healthcare professionals. Partnerships with multistakeholder such as community groups, public health and local leaders have been successful in improving individual and population health and health equity [1,2].

Developing rapport was a crucial component of providing patient-centered that ATs described in our study. This was implemented by ATs having conversations with their patients to build trust and then acting as a resource for their patient and their family. Furthermore, ATs were able to provide personalized care and continuous follow-up for their patients, especially for those that do not speak English as a first language or have difficulty with health literacy. The interventions implemented at the patient level by the secondary school ATs all demonstrate effective methods of positively impacting the delivery of health care and reduction of health disparities [6,7,40].

Our study demonstrated that ATs felt unprepared for their ATP to overcome many of the barriers associated with low SES patients and their SODH unless they were able to have a clinical site that exposed them to low SES patient-centered care. Many ATs spoke on these clinical sites as tremendously impactful and making them aware of the SODH related to low SES patients. They were able to learn from preceptors who knew how to navigate these challenges in low SES patient care through their experiences. Other ATs described that having awareness and knowledge of strategies to overcome these barriers through the classroom or clinical setting in their ATP could have helped them feel more prepared as compared to only being in a collegiate setting where their patient population does not have as many barriers to health care. This aligns with information from the Education Longitudinal Study demonstrating students from the most privileged backgrounds were more than three times as likely to be college athletes as those from disadvantaged backgrounds [41]. Having college athletics as the primary method of clinical education for MSAT students can hinder understanding of the impact of low SES on clinical care and how to navigate it.

Unfortunately, it is not as simple to ask ATP clinical coordinators to have low-SES secondary schools as clinical sites. As of 2015, only 37% of public secondary schools in the U.S. have full-time AT [42]. The presence of an employed AT on-site is negatively impacted by the median household income and percentage of free-reduced lunch students of school [12]. Barter et al. identified significant differences in public secondary school SES and AT services, with secondary schools of lower SES having less access to ATs and the care they provide [13]. Similarly, Robison et al. identified that in schools that employ an AT, schools in disadvantaged SES communities reported lower rates of contact frequencies for injury-related care, such as fewer AT room visit days/injury, fewer AT services/injury, and fewer AT services/AT room visit days [14]. Without an AT preceptor as a contact at a low SES school, it would make it difficult to provide clinical education that exposed AT students to the SDOH. This shows the need for ATP to integrate these concepts of the SDOH into their education programs through clinical case studies, patient simulations, understanding of health statistics, local community programs, legislations, health literacy and language barrier resources [13,32,43].

### 4.4. Limitations and Future Research

Research, particularly qualitative research, has inherent biases; however, the IPA and multiple-analyst triangulation with an external reviewer tries to minimize those biases by requiring consensus in the developing codebook at all 4 steps of the IPA process and implementing Yardley’s principles [24]. Our findings speak to the secondary school ATs’ perceptions, challenges, and strategies for navigating clinical care in their low SES patient populations; therefore, these findings cannot be generalized to other settings. We believe our work provides the foundation for future researchers to examine ATs’ perceptions, challenges, and strategies for navigating low SES clinical care in other settings, as well as the need for equitable AT care in every secondary school. Furthermore, future research should focus on AT-low SES patients’ perceptions of AT care and their navigations of their SES and SDOH challenges.

Lastly, participants volunteered for this study. Of these 12 participants, the majority were mainly white race/ethnicity and from public school settings. Although the 12 ATs were chosen at random out of the 139 ATs that volunteered, self-selection may have indicated certain assumptions and biases toward providing care to low SES patients.

## 5. Conclusions

Our study demonstrated that secondary school ATs are well-positioned to support and advocate for their low SES patients dealing with SDOH challenges to reduce health disparities. Our study makes evident the impact ATs have on low SES patient health care at the patient level, practice level, and community level. Despite being well-positioned, ATs were initially described in their careers to be ill-equipped from their ATP to navigate SES challenges as they delivered care, engaged in in-house patient care, and accessed the health care system, such as with referral and advanced imaging. ATs were able to lean on their clinical education from their ATP and accumulated experience as certified AT to provide a high standard of care. MSAT programs should emphasize classroom instruction on the SDOH, clinical education in low SES settings, when possible, low SES patient simulations, and collaboration with other healthcare professions to best prepare future generations of ATs in the secondary school setting.

## Figures and Tables

**Figure 1 ijerph-19-16709-f001:**
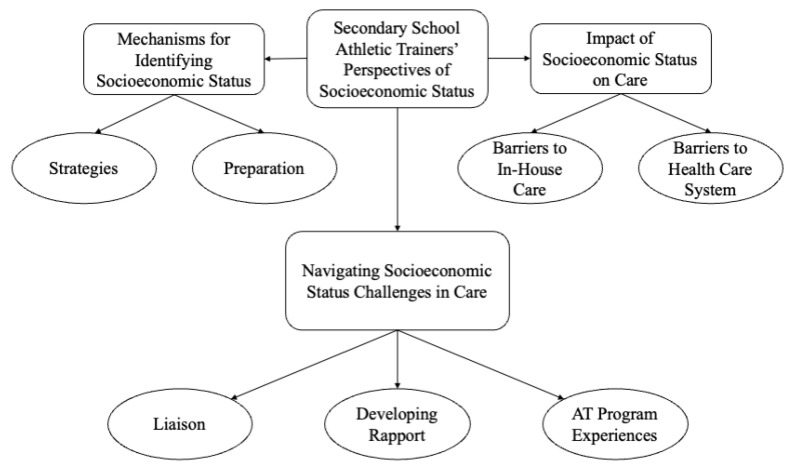
Secondary School Athletic Trainers’ Perspective of Socioeconomic Status Themes and Subthemes.

**Table 1 ijerph-19-16709-t001:** Interview Protocol *.

1. To begin, tell me about your background as an AT.
2. What is your definition of a low SES person?
3. Have you ever found yourself making assumptions about your patients because of their SES? Tell me more about that.
4. In your secondary school of employment, how do you know which student athletes are of low SES? At what time point do you know their SES? How did you learn about that information?
5. What experience, if any, do you have in providing care to patients who are of low SES? In what ways, in any, have those experiences changed how you view low SES patients?
6. In what ways, if any, does your experience in providing care to low SES patients affect your clinical decisions as an AT? Further probe with how this changes once the patient needs advanced care/AT needs to work alongside physician
7. In what ways, if any, does your experience in providing care to low SES patients affect your workload? Further probe: Does this mean more in house-care or a more conservative care plan?
8. What are, if any, the challenges you have faced in providing care to low SES student athletes? Further probe: How do these challenges make you feel?
9. When providing health care to low SES student athletes, what strategies do you feel have worked best for providing a high standard of care? Further probe: Why do you feel those strategies are successful?
10. What are the biggest barriers, if any, your low SES student athletes face when in the health care system?
11. In what ways, if any, did your ATP prepare you to provide care for low SES patient population?
12. Is there anything else you would like to share about providing care to low SES student athletes in the secondary school setting?

* Items are presented in their original format. Abbreviations: AT = athletic trainer, SES = socioeconomic status, ATP = athletic training program.

**Table 2 ijerph-19-16709-t002:** Participant Demographics.

Pseudonym	Sex	School Location	School Setting	Title 1 School	Highest Level of Education	Years of Clinical Experience	Race/Ethnicity
1	Male	Illinois	Private	No	Master’s	30	White
2	Male	Pennsylvania	Public	Yes	Master’s	3	White
3	Male	Pennsylvania	Private	No	Master’s	6	Hispanic, Latino, or Spanish Origin
4	Female	California	Public	No	Master’s	19	Hispanic, Latino, or Spanish Origin
5	Female	Kansas	Public	Yes	Master’s	11	White
6	Female	Arizona	Public	Yes	Master’s	28	White
7	Female	Pennsylvania	Public	Yes	Master’s	7	White
8	Female	Idaho	Public	Yes	Master’s	6	White
9	Female	Arizona	Public	Yes	Bachelor’s	3	White
10	Female	Indiana	Public	Yes	Bachelor’s	8	White
11	Female	Indiana	Public	Yes	Master’s	29	White
12	Female	Virginia	Public	No	Master’s	4	White

**Table 3 ijerph-19-16709-t003:** Mechanisms for Identifying Socioeconomic Status.

Category	Supporting Quotation
Strategies	“Because you got a kid that’s got a $50,000 souped out Jeep Cherokee driving in and you got another one, that’s got a small convertible that you can hear the muffler rattling or the kids that are walking home. And it’s not walking because it’s close, they’re just walking because that’s the transportation.”—P1
	“My biggest thing is when I can look at their physicals and I see that their insurance is either Medicaid or no insurance, or the parents will sometimes disclose to me, “we don’t have insurance, we can’t afford to go to physical therapy, can we do our rehab with you?” So, it’s a combination of that. Either seeing it on the documents or the parents of the kids disclosing it to me.”—P10
	“I have conversations with coaches, teachers, my athletic director, et cetera, at the beginning of the school year, particularly for the incoming students, because the ones who are sophomores, juniors, seniors, I usually have had before, and I know them. So, I usually have conversations with those other adults about the students, and that’s typically where I get my information on their socioeconomic status.”—P8
	“Most of the time the kids are forthcoming. “Like, yeah, I live in Chula Vista, which is really far away or yeah, my parents are working two jobs, or I have five siblings and my parents are working two jobs.” So, it’s kind of what other information I can gather from them without directly asking.”—P4
Preparation	“I probably got a better eye opener in my teaching education program because I was assigned to a low-income elementary school. I still remember the teacher that I went with had a lot of years of experience. She said, “This is the best meal they’re going to get.” It really explained to me that we have like 60% of our students here are very low income that come in. So, I actually got more from my student teaching part than I ever got from my athletic training part. I think that’s because you don’t have those real experiences....”—P1
	“I think it was more so our clinical education that allowed me to get that understanding because my very first clinical site was actually at a very rural high school. So that was kind of my first eye-opening experience of, these kids aren’t coming from a lot. These kids are getting a free and reduced lunch and sometimes that’s their only means of a meal for that day.”—P12
	“Oh, it’s a rude awakening for me. I would have been prepared if I had one of my clinical rotations at this low SES site or at a similar site. I would have been like, “Okay, I can recognize which athletes are low SES and which athletes aren’t.”—P2

**Table 4 ijerph-19-16709-t004:** Impact of SES on Care.

Category	Supporting Quotation
Barriers to in-house care	“We don’t have that specialty store that some places have. Rehab stuff we’re kind of bare on. We don’t have a lot of the fancy stuff. The STIM machine came, so that helped, but it’s still kind of just getting creative with what we have.”—P12
	“Cross country training shoes costs $80 and $180 per pair these kids can’t afford it, but they still want to be a part of the team. I’ve seen cross country kids that have come in with holes in their shoes and that’s the only pair of shoes that they have. So, they wear them to school all day long and then they try to go do a five-mile run and they come in and you wonder why their body is hurting.”—P5
	“We have a couple of kids whose parents speak very little English or are only Spanish speaking. So, I have to use the child as a translator, which I’ve learned now is not best practice. When I’m trying to convey information to the parent on how to best care for their child and using the child who we’re talking about as the translator, to me, doesn’t feel like it’s the right thing to do because you shouldn’t be using children to translate medical things. They may not understand how to appropriately translate what’s going on. My fear is that what I’m saying isn’t making it to the parents in a way that they can comprehend and understand and make an appropriate decision.”—P10
	“Sometimes it’s the kids just being non-compliant with daily screenings and rehab. Sometimes it’s the families being non-compliant. “If little Bobby has a head injury and can’t practice, well he’s going to come home and babysit his siblings. So, he doesn’t need to see you. He’ll come in eventually when he feels better.”—P6
Barriers to health care system	“I think they tend to not get as good of care just based on where they geographically lived. The hospitals and clinics there are not as good as where a lot of my highest SES students live.”—P3
	“It’s about 20 plus miles to the nearest hospital, the nearest specialist, really the nearest health clinic essentially. So, that’s a struggle that I have to deal with where I can’t just necessarily go, “I think this person should be referred. Okay, great. I’m going to refer them.” Do they have the ability to go see this doctor if I was to refer them?”—P8
	“I may need to explain to the parents that their kid may not get the MRI in two days like another kid on the team did. You know, kids talk amongst themselves. Obviously, I’m not going to share information, but I let them know that it may not be tomorrow that you get your MRI. It may be another week or so. We can help you, but the process may be a little bit different for you.”—P10
	“My gut says he probably broke his scaphoid. I would’ve loved to have him get X-rayed, but the mother fought me tooth and nail. Undocumented immigrant, didn’t want to be in the system, didn’t want all that.”—P6

**Table 5 ijerph-19-16709-t005:** Navigating Socioeconomic Status Challenges in Care.

Category	Supporting Quotation
Liaison	“There was one physician practice in town, and I had an amazing relationship with her. It was really easy to say, “Hey, so-and-so got hurt”, and she would be like, “I know that family, tell him that I’ll see them in a week if I need to see him”, or she’d be like, “I can go see him.” Sometimes it’d be like, “You know what, I’ll stop by their house tonight”, and she’d do it for free.”—P7
	“In many instances I play the gatekeeper for medical care. Meaning, if it was something we could take care of in house I would communicate with the parents, and they were happy with it. If it was something that I knew was out of my hands, then I would go through the resources I had available, work with our dropout prevention coordinator, refugee services on campus to get these kids seen. In a lot of instances, if the parents bought into it, you could get them the access, the state health insurance.”—P6
	“My orthopedist is awesome, and I can send my kids to him if I really need it. But there are certain places and there are certain doctors that I just won’t send my kids to, because I know that first of all, they will be discriminated against because of what they look like and where they live, and they won’t take them because they wouldn’t be able to afford it.”—P9
	“I’m more prepared to be able to help them with braces or crutches or things that they need. I have former athletes whose parents are cleaning out when they go away to college. And they say, “can you use this cryo cuffs? Yes. Can you use these crutches? Yes. Can you use these ankle braces? Yes.” Because I always know I have kids that can’t afford them, and we’ll be grateful for them. Especially if they’ve had previous injuries and stuff like that. I just have never turned down a hand.”—P11
Developing Rapport	“They need to understand that as an athletic trainer and a teacher that I’m not going to go running around town to tell them what you tell me. I’m a resource that you can use. I can help you. I don’t have all the answers, but at least I think being in a community long enough, I know where I can tell them to go to get the answers, to help them.”—P1
	“I don’t think you ever really realize a situation or what a kid is going through until you actually sit down and have that conversation with them. Being able to build that trust is huge, but it takes time. They’re not just going to automatically trust you right off the bat because of the situations that these kids come from or go through.”—P12
	“I don’t think they get follow up care or personalized medical care or they get charged a bill and they get stressed about it. Instead, I say, “let’s try to avoid that. And then if we need to, we’ll do it, but I’m still going to follow up with you regardless of how this case turns out.”—P10
	“I try talk to them about what I do, what I can do for their kids as an athletic trainer, and I really stress that I’m there to take care of their kids. There’s no ulterior motive or anything like that. Some of them, especially with my African American athletes and parents, have disclosed to me that they really just don’t trust doctors, or just healthcare in general, because they have been mistreated in the past.”– P9
ATP Experiences	“I think we just assumed people had the resources available to them to just do the gold standard of care, which is not the real world. Most of my experiences were in the college setting so everything was kind of in-house and taken care of. What I remember from my first rotation at that high school and what I’ve even seen now is that there are so many factors that come into the care people actually get. I don’t think we ever talked about the fear undocumented people might have going to a doctor’s office or language barriers, access to interpreters, things like that.”—P3
	“My only exposure to diverse populations was at this one site and I think that’s what drew me in. There wasn’t really a lot of education on certain populations or how to go about if the lower SES student can’t afford to go to get an x-ray or something like that. So, it was very eye-opening once I got to that one rotation and then once I was an independent clinician at my school.”—P9
	“The collegiate level for my undergrad just focused on the athletes who were on our campus. So, every athlete is almost equal at that point because they’re living on campus or nearby campus. So, we never really explored or dove into any differences. Everybody can see doctor so-and-so. All had free reign to student health. So there never was an issue of socioeconomic status.”—P4

## Data Availability

Not applicable.

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
