# Peer review of "Secondary School Athletic Trainers’ Navigation of Patient Socioeconomic Status Challenges in Care: A Qualitative Study"

_ijerph, 2022, doi:10.3390/ijerph192416709_

Round 1
Reviewer 1 Report
This is a timely and important paper addressing the important and unique role of secondary school athletic trainers' and the barriers they experience working with students from high poverty communities. The introduction and methods are detailed and well written. The results are nicely organized and discussion well written. My only concern is that the quotes are far more compelling than the summary. Specifically, the summary focuses on well known concerns for lack of health care and other resources for children living in poverty. What is more interesting in the quotes is how that athletic trainers navigated these issues. This is a relatively easy fix for the authors. My suggestion is to focus less on the context -- which is not new news -- and more on the innovative strategies -- and struggles -- of this under-appreciated workforce.
In summary, I congratulate the authors on a well-constructed and important study. I suggest with a bit of editing, the paper will provide an important and new focus on the unheralded work of athletic directors to support children's development.
Author Response
Manuscript ID: ijerph-1952653
Reviewer 1:
This is a timely and important paper addressing the important and unique role of secondary school athletic trainers' and the barriers they experience working with students from high poverty communities. The introduction and methods are detailed and well written. The results are nicely organized and discussion well written. My only concern is that the quotes are far more compelling than the summary. Specifically, the summary focuses on well-known concerns for lack of health care and other resources for children living in poverty. What is more interesting in the quotes is how that athletic trainers navigated these issues. This is a relatively easy fix for the authors. My suggestion is to focus less on the context -- which is not new news -- and more on the innovative strategies -- and struggles -- of this under-appreciated workforce.
In summary, I congratulate the authors on a well-constructed and important study. I suggest with a bit of editing, the paper will provide an important and new focus on the unheralded work of athletic directors to support children's development.
Response: Thank you for this valuable feedback. The authors have made edits to focus on the strategies and struggles of secondary school ATs as they navigate their patient’s SES barriers to health care.
Reviewer 2 Report
Your idea of getting information about secondary school Atletic Trainers (AT) experiences with providing care to the low Social Economic Status (SES) patient population is a good starting point to get new information about other forms of access to care for a low SES population.
I don't find table 1
The qualitative methodology based on questionaires and using interpretative phenomenological analysis research approach allowed to obtain three main relevant themes and characterized them. The manuscript referes to a type of profession that doesn't work in the same setting in every country, for instance in some countries sports teachers are not healthcare personel, nevertheless the manuscript is worth reading
Author Response
Manuscript ID: ijerph-1952653
Reviewer 2:
Your idea of getting information about secondary school Athletic Trainers (AT) experiences with providing care to the low Social Economic Status (SES) patient population is a good starting point to get new information about other forms of access to care for a low SES population.
I don't find table 1. The qualitative methodology based on questionnaires and using interpretative phenomenological analysis research approach allowed to obtain three main relevant themes and characterized them. The manuscript refers to a type of profession that doesn't work in the same setting in every country, for instance in some countries sports teachers are not healthcare personnel, nevertheless the manuscript is worth reading.
Response: Thank you so much for your feedback. The authors greatly appreciate your support. I have attached table 1 to this response and the manuscript revision. We also acknowledge that ATs are mainly located in the United States but countries like Canada have Athletic Therapist which are comparable. Although we also do acknowledge we cannot generalize to other countries due to the SES and Social Determinants of Health being different by country and region.
